# Patch Diffusion: Faster and More Data-Efficient Training of Diffusion Models

**Zhendong Wang**[1,2], **Yifan Jiang**[1], **Huangjie Zheng**[1,2], **Peihao Wang**[1], **Pengcheng He**[2],
**Zhangyang Wang**[1], **Weizhu Chen**[2], **and Mingyuan Zhou**[1]
[1]The University of Texas at Austin, [2]Microsoft Azure AI

## Abstract

Diffusion models are powerful, but they require a lot of time and data to train. We propose *Patch Diffusion*, a generic patch-wise training framework, to significantly reduce the training time costs while improving data efficiency, which thus helps democratize diffusion model training to broader users. At the core of our innovations is a new conditional score function at the patch level, where the *patch location* in the original image is included as additional coordinate channels, while the *patch size* is randomized and diversified throughout training to encode the cross-region dependency at multiple scales. Sampling with our method is as easy as in the original diffusion model. Through Patch Diffusion, we could achieve $\geq 2\times$ faster training, while maintaining comparable or better generation quality. Patch Diffusion meanwhile improves the performance of diffusion models trained on relatively small datasets, *e.g.*, as few as 5,000 images to train from scratch. We achieve outstanding FID scores in line with state-of-the-art benchmarks: 1.77 on CelebA-64×64, 1.93 on AFHQv2-Wild-64×64, and 2.72 on ImageNet-256×256. We share our code and pre-trained models at https://github.com/Zhendong-Wang/Patch-Diffusion.

## 1 Introduction

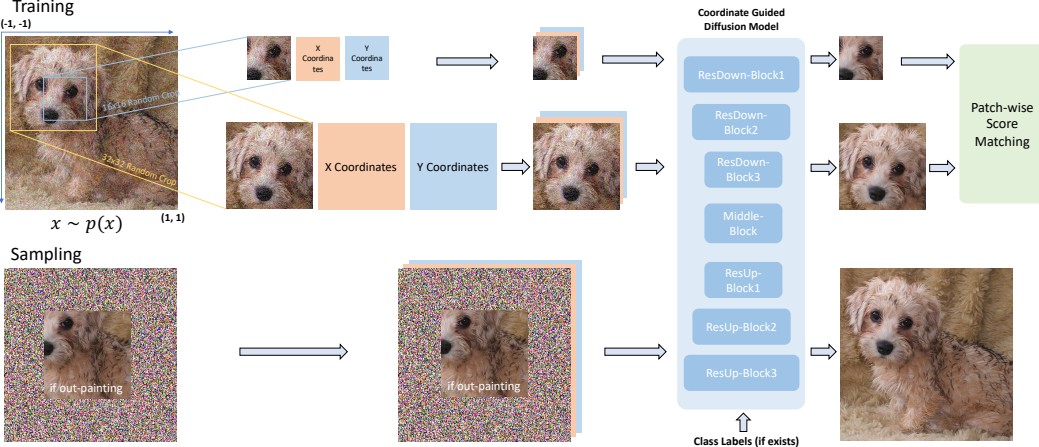

Figure 1: Illustration of Patch Diffusion on training and sampling.

Diffusion models [50, 15] have become the new *de facto* generative AI (GenAI) models. Song et al. [54] unite the diffusion models into the framework of score-based generative models [17, 57, 52, 53].

37th Conference on Neural Information Processing Systems (NeurIPS 2023).

Diffusion models have found great success in a wide range of applications, including unconditional image synthesis [15, 51, 40, 22], text-to-image generation [39, 47, 44, 45, 61], audio generation [24, 43], uncertainty quantification [12], and reinforcement learning [19, 59].

Although diffusion models are stable to train and powerful in capturing distributions, they are notoriously slow in generation due to the need to traverse the reverse diffusion chain, which involves going through the same U-Net-based generator network hundreds or even thousands of times [51, 60]. As such, there is great interest in improving the inference speed of diffusion models, leading to many improved samplers such as DDIM [51], TDPM [67], DPM-Solver [35, 36], and EDM-Sampling [22].

However, diffusion models are also notoriously expensive and data-hungry to train. They require large datasets and many iterations to capture the high-dimensional and complex data distributions. For instance, training DDPM [15] on eight V100 GPUs for the LSUN-Bedroom [63] dataset at resolutions of $64 \times 64$ and $256 \times 256$ takes approximately four days and over two weeks, respectively. A state-of-the-art diffusion model in Dhariwal & Nichol [10] consumes 150-1000 V100 GPU days to produce high-quality samples. The training costs grow exponentially as the resolution and diversity of the target data increase. Moreover, the best-performing models [47, 45] rely on billion-level image datasets such as OpenImages [26] and LAION [48], which are not easily accessible or scalable.

While some most exciting GenAI results are arguably accomplished by training diffusion models with enormous computational power and data (often owned or led by large corporations), the **prohibitive time and data scale** required to train competitive diffusion models have presented a critical bottleneck for democratizing this GenAI workhorse technology to the broader research community who generally lack access to such high-end privileged resources. This problem of **democratizing diffusion model training** is becoming pressing, yet largely overlooked so far.

To democratize diffusion model training, we propose patch-wise diffusion training (**Patch Diffusion**), a plug-and-play training technique that is agnostic to any choice of UNet [15, 22] architecture, sampler, noise schedule, and so on. Instead of learning the score function on the full-size image for each pixel, we propose to learn a conditional score function on image patches, where both *patch location* in the original image and *patch size* are the conditions. Training on patches instead of full images significantly reduces the computational burden per iteration. To incorporate the conditions of patch locations, we construct a pixel-level coordinate system and encode the patch location information as additional coordinate channels, which are concatenated with the original image channels as the input for diffusion models. We further propose diversifying the patch sizes in a progressive or stochastic schedule throughout training, to capture the cross-region dependency at multiple scales.

Patch Diffusion aims to significantly reduce the training time costs while improving the data efficiency of diffusion models. Sampling with our method is as easy as it in the original diffusion model: we compute and parameterize the full coordinates with respect to the original image, concatenate them with the sampled noise, and then reverse the diffusion chain to collect samples, as illustrated in Figure 1. Through Patch Diffusion, we could achieve $\geq \mathbf{2\times}$ faster training, while maintaining comparable or better generation quality. We also observe that Patch Diffusion improves the performance of diffusion models trained on relatively small datasets, *e.g.*, clearly superior generation results when training with as few as 5,000 images from scratch. We summarize our main contributions as follows:

- The first patch-level training framework, generally applicable to diffusion models, that targets saving training time and data costs.
- Novel strategies of patch coordinate conditioning and patch size conditioning/scheduling, to balance training efficiency and effective global structure encoding.
- Competitive results while generally halving the training time, and notable performance gains at small training data regimes.

## 2   Related Work

**Preliminaries for diffusion models.**   Diffusion probabilistic models [50, 15] first construct a forward process that injects noise into data distribution, then reverse the forward process to reconstruct it. The forward process iteratively applies noise to the given images, whereas the reverse process iteratively denoises a noisy observation. After the origin of the diffusion model, many efforts have been made to apply it to various downstream tasks. Ho et al. [16] and Vahdat et al. [56] propose a hierarchical architecture to stabilize the training process of diffusion models and further mitigate

memory cost issues. Dall·E-2 [44] first introduces the diffusion model to the text-to-image generative task and achieves remarkable success. Later, Saharia et al. [47] found that increasing the parameter of the language model improves both sample fidelity and image-text alignment much more than increasing the size of the image model.

**Data-efficient training in generative models.** Prior to the study of diffusion models, many explorations have been conducted on limited data or even the few-shot setting of generative model's training schemes, $e.g.$, Generative Adversarial Networks (GANs). To avoid expensive data collection, Zhao et al. [66] propose differentiable augmentation on both the generator and discriminator of GANs and achieve acceptable performance with only 10% data. Karras et al. [21] apply adaptive augmentation strategy for the setting of limited data to prevent information leakage. Wang et al. [60] propose to train GANs with an adaptive forward diffusion process that injects different levels of noise during training, significantly improving GANs' performance in small datasets. Chen et al. [6] discover the lottery ticket mask of a specific GAN architecture that allows further reducing the training data. Tseng et al. [55] propose to add regularization on top of the existing data augmentation strategy and stabilize the brittle training process of adversarial learning. Others explore the domain adaptation of GAN frameworks when only a few data from target domains are available, $e.g.$, by using elastic weight consolidation [30] or cross-domain correspondence [42].

Meanwhile, recent works also start exploring the few-shot adaptation of diffusion models [37]. DreamBooth [46] uses four frontal images of a specific subject to finetune a pre-trained diffusion model and additionally set a special indicator in the prompt, where the obtained model is able to generate various examples with the same identity. Later, Zhang et al. [65] reduce the number of required images to only a single example when finetuning the diffusion model, but still generate visually impressive photos given various prompt inputs. However, the above works do not fully explore the setting of training a diffusion model from scratch. Although a few single-image diffusion algorithms [58, 41, 25] only need a single training image, they can only produce variations of the same image and hence focus drastically differently compared to our proposed method.

**Resource-efficient training in generative models.** To address the issue of mode collapse given limited training data, Lin et al. [32] propose Anycost-GAN that is able to generate examples with different levels of cost during inference. However, their proposed techniques do not help save the training cost, and may even take a longer time to optimize the SuperNet [4]. Coco-GAN [31] firstly introduces the patch-wise training scheme on GAN frameworks. However, since their discriminator still needs to combine several generated patches together, its effectiveness on memory saving is not distinguishable. Lee et al. [27] further apply the patch-wise training scheme to the INR-GAN [49] framework. Although their approach is able to save GPU memory cost, it will hurt the quality of generated examples ($e.g.$, increasing FID on the FFHQ dataset from 8.51 to 24.38). In the meantime, since the low-resolution images are generally more accessible compared to high-resolution ones, Chai et al. [5] adopt any-resolution framework which is able to blend data with different resolutions together to feed the training of GANs, and therefore reduce the partition of the high-resolution part. Despite its superiority of data efficiency, it does not particularly discuss the training costs and the effectiveness of its methodology remains unknown and unexplored for diffusion models.

A handful of works try to tackle the issue of huge training and inference costs in diffusion models. Rombach et al. [45] perform diffusion in a latent space instead of the pixel space, which largely reduces the training and inference cost. Other works [35, 36, 51, 2, 1] study the fast sampling strategy at the inference stage, which does not directly help speed up the training process. Our proposed approach is orthogonal to the aforementioned works and can serve as a plug-and-play module.

Simultaneously with our research efforts, Ding et. al. [11] concurrently developed another model of patch-based denoising diffusion. Their model excels in memory-efficient high-resolution image synthesis while avoiding the introduction of boundary artifacts. Central to their methodology is the introduction of a novel feature collage technique, achieved through a systematic window-sliding process, which effectively enforces spatial consistency.

## 3 Patch Diffusion Training

We first briefly review diffusion models from the perspective of score-based generative modeling and then present our method in details. Suppose we are given a dataset $\{\boldsymbol{x}_n\}_{n=1}^N$, where each datapoint is independently drawn from an underlying data distribution $p(\boldsymbol{x})$. Following Song et al.

[54], we consider a family of perturbed distributions $p_\sigma(\tilde{\boldsymbol{x}} \,|\, \boldsymbol{x}) = \mathcal{N}(\tilde{\boldsymbol{x}}; \boldsymbol{x}, \sigma\boldsymbol{I})$ obtained by adding independent, and identically distributed (i.i.d.) Gaussian noise with standard deviation $\sigma$ to the data. With a sequence of positive noise scales $\sigma_{\min} = \sigma_0 < \cdots < \sigma_t < \cdots < \sigma_T = \sigma_{\max}$, a forward diffusion process, $p_{\sigma_t}(\boldsymbol{x})$ for $t = 1, \ldots, T$, is defined given that $\sigma_{\min}$ is small enough such that $p_{\sigma_{\min}}(\boldsymbol{x}) \approx p(\boldsymbol{x})$ and $\sigma_{\max}$ is large enough such that $p_{\sigma_{\max}}(\boldsymbol{x}) \approx \mathcal{N}(\boldsymbol{0}, \sigma_{\max}^2\boldsymbol{I})$. The target of diffusion models is to learn how to reverse the chain and restore the data distribution $p(\boldsymbol{x})$.

Further generalizing to an infinite number of noise scales [54], $T \to \infty$, the forward diffusion process could be characterized by a stochastic differential equation (SDE) and further converted to an ordinary differential equation (ODE). The closed form of the reverse SDE is given by

$$d\boldsymbol{x} = \left[\boldsymbol{f}(\boldsymbol{x}, t) - g^2(t)\nabla_{\boldsymbol{x}} \log p_{\sigma_t}(\boldsymbol{x})\right] dt + g(t)d\boldsymbol{w}, \tag{1}$$

where $\boldsymbol{f}(\cdot, t) : \mathbb{R}^d \to \mathbb{R}^d$ is a vector-valued function called the drift coefficient, $g(t) \in \mathbb{R}$ is a real-valued function called the diffusion coefficient, $dt$ represents a negative infinitesimal time step, $\boldsymbol{w}$ denotes a standard Brownian motion, and $d\boldsymbol{w}$ can be viewed as infinitesimal white noise. The corresponding ODE of the reverse SDE is named probability flow ODE [54], given by

$$d\boldsymbol{x} = \left[\boldsymbol{f}(\boldsymbol{x}, t) - 0.5g^2(t)\nabla_{\boldsymbol{x}} \log p_{\sigma_t}(\boldsymbol{x})\right] dt. \tag{2}$$

As shown in Equations (1) and (2), solving the reverse SDE or ODE requires us to know both the terminal distribution $p_{\sigma_T}(\boldsymbol{x})$ and score function $\nabla_{\boldsymbol{x}} \log p_{\sigma_t}(\boldsymbol{x})$. By design, the former $p_{\sigma_T}(\boldsymbol{x})$ is close to a white Gaussian noise distribution. The analytical form of $\nabla_{\boldsymbol{x}} \log p_{\sigma_t}(\boldsymbol{x})$ is generally intractable, and hence we learn a function $s_{\boldsymbol{\theta}}(\boldsymbol{x}, \sigma_t)$ parameterized by a neural network to estimate its values. Denoising score matching [57, 51] is currently the most popular way of estimating score functions applied in diffusion models. After learning the estimated score function $s_{\boldsymbol{\theta}}(\boldsymbol{x}, \sigma_t)$, we can obtain an estimated reverse SDE or ODE to collect data samples from the estimated data distribution.

Next, we introduce our patch diffusion training in three subsections. First, we propose to conduct conditional score matching on randomly cropped image patches, with the patch location and patch size as conditions, to improve the efficiency of learning scores. Second, we introduce pixel coordinate systems to provide better guidance on patch-level score matching. Then, we show by using our method, for each reverse step, we could do sampling globally as easily as the original diffusion models, without the need to explicitly sample separate local patches and merge them afterwards.

## 3.1 Patch-wise Score Matching

Following denoising score-matching in Karras et al. [22], we build a denoiser, $D_{\boldsymbol{\theta}}(\boldsymbol{x}; \sigma_t)$, that minimizes the expected $L_2$ denoising error for samples drawn from data distribution $p(\boldsymbol{x})$ independently for any $\sigma_t$:

$$\mathbb{E}_{\boldsymbol{x} \sim p(\boldsymbol{x})}\mathbb{E}_{\boldsymbol{\epsilon} \sim \mathcal{N}(\boldsymbol{0}, \sigma_t^2\boldsymbol{I})}||D_{\boldsymbol{\theta}}(\boldsymbol{x} + \boldsymbol{\epsilon}; \sigma_t) - \boldsymbol{x}||_2^2.$$

Thus we have the score function as

$$s_{\boldsymbol{\theta}}(\boldsymbol{x}, \sigma_t) = (D_{\boldsymbol{\theta}}(\boldsymbol{x}; \sigma_t) - \boldsymbol{x})/\sigma_t^2. \tag{3}$$

Instead of conducting the score matching on the full images, we propose to learn the score function on random-size patches. As shown in Figure 1, for any $\boldsymbol{x} \sim p(\boldsymbol{x})$, we first randomly crop small patches $\boldsymbol{x}_{i,j,s}$, where we use $(i, j)$, the left-upper corner pixel coordinates, to locate each image patch, and $s$ to denote the patch size, $e.g.$, $s = 16$. We conduct denoising score matching on image patches with the corresponding patch locations and sizes as the conditions, expressed as

$$\mathbb{E}_{\boldsymbol{x} \sim p(\boldsymbol{x}), \boldsymbol{\epsilon} \sim \mathcal{N}(\boldsymbol{0}, \sigma_t^2\boldsymbol{I}), (i,j,s) \sim \mathcal{U}}||D_{\boldsymbol{\theta}}(\tilde{\boldsymbol{x}}_{i,j,s}; \sigma_t, i, j, s) - \boldsymbol{x}_{i,j,s}||_2^2, \tag{4}$$

where $\tilde{\boldsymbol{x}}_{i,j,s} = \boldsymbol{x}_{i,j,s} + \boldsymbol{\epsilon}$ and $\mathcal{U}$ denotes the uniform distribution on the corresponding value range, $e.g.$, $i \sim [-1, 1]$. Then, our conditional score function derived from Equation (3), $s_{\boldsymbol{\theta}}(\boldsymbol{x}, \sigma_t, i, j, s)$, is defined on each local patch. We learn the scores for pixels within each image patch conditioning on its location and patch size.

With Equation (4), the training speed is significantly boosted due to the use of small local patches. However, the challenge now lies in that the score function $s_{\boldsymbol{\theta}}(\boldsymbol{x}, \sigma_t, i, j, s)$ has only seen local patches and may have not captured the global cross-region dependency between local patches, in other words, the learned scores from nearby patches should form a coherent score map to induce coherent image sampling. To resolve this issue, we propose two strategies: 1) random patch sizes and 2) involving

a small ratio of full-size images. Our training patch size is sampled from a mixture of small and large patch sizes, and then the cropped large patch could be seen as a sequence of small patches. In this way, the score function $s_\theta$ learns how to unite the scores from small patches to form a coherent score map when training on the large patches. To ensure the reverse diffusion converges towards the original data distribution, in some iterations during training, full-size images are required to be seen.

We further provide a theoretical interpretation in Appendix A to help understand our method, and conduct a detailed empirical study in Section 4.1 to show the impact of the ratio of full images.

## 3.2 Progressive and Stochastic Patch Size Scheduling

To strengthen the awareness of score function in cross-region dependency, we are motivated to propose patch-size scheduling. Denote the ratio of iterations that takes as input the full-size images as $p$, and the original image resolution as $R$. We propose to use the patch options, as follows

$$s \sim p_s := \begin{cases} p & \text{when } s = R, \\ \frac{3}{5}(1-p) & \text{when } s = R//2, \\ \frac{2}{5}(1-p) & \text{when } s = R//4. \end{cases} \quad (5)$$

Two patch-size schedulings could be considered. 1) Stochastic: During training, we randomly sample $s \sim p_s$ for each mini-batch with the probability mass function defined in Equation (5). 2) Progressive: We train our conditional score function from small patches to large patches. In the first $\frac{2}{5}(1-p)$ training iterations, we fix the patch size as $s = R//4$, while in the second $\frac{3}{5}(1-p)$ training iterations, $s = R//2$ is applied. Finally, we train the models on full-size images for $p$ ratio of the total iterations.

Empirically, we find that $p = 0.5$ with stochastic scheduling reaches a sweet point in the trade-off between training efficiency and generation quality, as shown in Figure 3.

One natural question is why the denoiser $D_\theta$ could handle image patches of varying sizes. We note the UNet [15, 22] architecture is fully designed with convolutional layers, and the convolutional filters are capable of handling any resolution images by moving themselves around the inputs. Hence, our patch diffusion training could be regarded as a plug-and-play training technique for any UNet-based diffusion models. The flexibility of UNet on resolutions also makes our sampling easy and fast.

## 3.3 Conditional Coordinates for Patch Location

Motivated by COCO-GAN [31], to further incorporate and simplify the conditions of patch locations in the score function, we build up a pixel-level coordinate system. We normalize the pixel coordinate values to $[-1, 1]$ with respect to the original image resolution, by setting the upper left corner of the image as $(-1, -1)$ while the bottom right corner as $(1, 1)$.

As shown in Figure 1, for any image patch $x_{i,j,s}$, we extract its $i$ and $j$ pixel coordinates as two additional channels. For each training batch, we independently randomly crop each data sample with a sampled patch size for the batch and extract its corresponding coordinate channels. We concatenate the two coordinate channels with the original image channels to form the input of our denoiser $D_\theta$. When computing the loss defined in Equation (4), we ignore the reconstructed coordinate channels and only minimize the loss on the image channels.

The pixel-level coordinate system together with random patch size could be seen as a kind of data augmentation method. For example, for an image with resolution $64 \times 64$, with patch size $s = 16$, we could have $(64 - 16 + 1)^2 = 2401$ possible patches with different locations specified. Hence, we believe training diffusion models on patch-wise would help the data efficiency of diffusion models, in other words, with our method, diffusion models could perform better on small datasets. We validate this hypothesis through experiments in Section 4.4.

## 3.4 Sampling

By utilizing our coordinate system and the UNet, we are able to easily accomplish the reverse sampling defined in either Equation (1) or Equation (2). As we have shown in Figure 1, we compute and parameterize the coordinates for the full image, and concatenate them together with the image sample from last step as coordinate conditions at each reverse iteration. We abandon the reconstruction output of coordinate channels at each reverse iteration.

We provide a theoretical interpretation of Patch Diffusion Training in Appendix A.

## 4    Experiments

We conduct five sets of experiments to validate our patch diffusion training method. In the first subsection, we conduct an ablation study on what impacts the performance of our method. In the second subsection, we compare our method with its backbone model and other state-of-the-art diffusion model baselines on commonly-used benchmark datasets. Thirdly, we show that our method could also help improve the efficiency of finetuning large-scale pretrained models. Then, we show that patch diffusion models could achieve better generation quality on typical small datasets. Finally, we evaluate the out-painting capability of patch diffusion models.

**Datasets.**    Following previous works [22, 8, 23], we select CelebA ($\sim$200k images) [34], FFHQ (70k images) [20], LSUN ($\sim$200k images) [63], and ImageNet ($\sim$1.2 million images) [9] as our large datasets, and AFHQv2-Cat/Dog/Wild ($\sim$5k images in each of them) [7] as our small datasets.

**Evaluation protocol.**    We measure image generation quality using Fréchet Inception Distance (FID) [13]. Following Karras et al. [20, 22], we measure FID using 50k generated samples, with the full training set used as reference. We use the number of real images shown to the diffusion models to measure our training duration [22]. Unless specified otherwise, all models are trained with a duration of 200 million images to ensure convergence (these trained with longer or shorter durations are specified in table captions). For the sampling time, we use the number of function evaluations (NFE) as a measure.

**Implementations.**    We implement our Patch Diffusion on top of the current state-of-the-art Unet-based diffusion model EDM-DDPM++ [22] and EDM-ADM [22]. EDM-DDPM++ is our default backbone model for training low-resolution ($64\times64$) datasets, while EDM-ADM coupling with Stable Diffusion [45] latent en/decoders is our backbone model for training high-resolution ($256\times256$) datasets. We inherit the hyperparameter settings and the UNet architecture from Karras et al. [22]. We implement random cropping independently on each data point in the same batch, and the corresponding pixel coordinates are concatenated with the image channels to form the input for the UNet denoiser. When computing the diffusion loss at any $\sigma$ level, we ignore the reconstructed pixel coordinates. We adopt the EDM-Sampling strategy with 50 deterministic reverse steps in the inference stage, for both the baseline model and ours.

### 4.1    Ablation study

We provide ablation studies to investigate what may impact the performance of Patch Diffusion.

**Impact of full-size images.**    We first study the main effect of $p$, the ratio of full-size images during training, regarding the quality of generated examples and the training cost. We conduct experiments on a $p$ gird, $[0.0, 0.1, 0.25, 0.5, 0.75, 1.0]$. Note here when $p = 0.0$, only patch images and their pixel coordinates are available during the training process, and when $p = 1.0$, patch training becomes the same as the standard training procedure of diffusion models. We train all models on 16 Nvidia V100 GPUs with a batch size of 512 for a duration of 200 million images.

We show the results in Figures 2 and 3. In the extreme case $p = 0.0$, the conditional score function is only trained on local patches, $e.g.$, $16 \times 16$ and $32 \times 32$ image patches, without the knowledge of how the full-size images look like. The FID is reasonably unsatisfactory while we still observe some coherently sampled images, such as the third and fourth faces shown in the 1st row of Figure 3. This observation validates our idea that using a mixture of large and small patch sizes could help the conditional score function to capture cross-region dependency. The learned small patch scores are guided by large patch score matching to form a coherent score mapping. Then, when $p$ is greater than 0, even if $p$ is as small as 0.1, the generation quality is dramatically improved in terms of FID. We reason this as that the involvement of full-size images provides the global score for patch-wise score matching, even though only a small ratio of the global score could guide the score function to learn towards it. As the $p$ increases, we could see the FID also becomes better, and we hypothesize this is because the conditional score function converges better towards the local minimum due to more global score guidance. However, the improvement of FID gained from increasing $p$ is not unlimited. We also observe that when $p$ is large enough, such as $p = 0.75$, the FID values converge

Figure 2: FID results on CelebA-64×64 with different $p$ values.

| Metric | p=0.0 | p=0.1 | p=0.25 | p=0.5 | p=0.75 | p=1.0 |
|---|---|---|---|---|---|---|
| FID | 14.51 | 3.05 | 2.10 | 1.77 | **1.65** | 1.66 |
| Training Time (in hrs) | 13.6 | 20.1 | 22.5 | 24.6 | 42.7 | 48.5 |

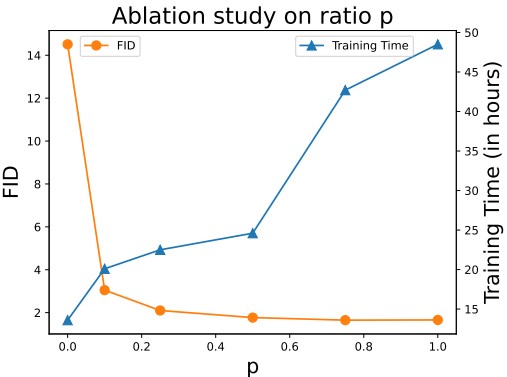

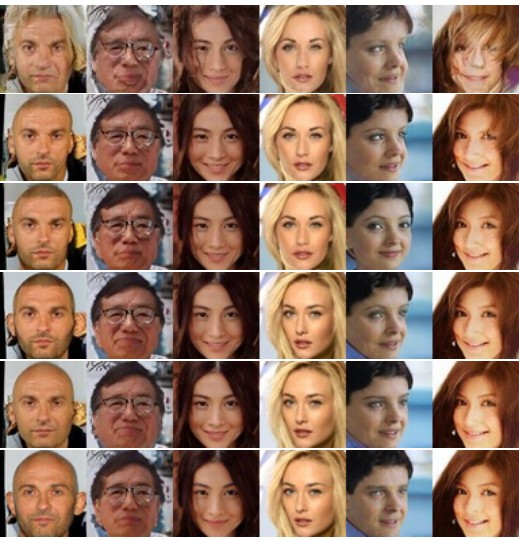

Figure 3: Ablation study on dataset CelebA-64×64 for the ratio of full-size images $p$. We generate from the trained models with different $p$, and show the generated images for a fixed set of seeds in the $p$ increasing order from top to bottom.

to the minimum level, which indicates that sparsely using full-size images during training could be sufficient to guide the conditional score-matching to converge.

On the other hand, larger $p$ means more training cost and longer training time. Hence, we pick the sweet point shown on the line plot, $p = 0.5$, which provides a good trade-off between generation quality and training efficiency, for our following experiments.

**Patch size scheduling.** Further, we investigate the impact of patch size scheduling, a stochastic or progressive way. Note usually the training of diffusion models applies learning rate decay. This may hurt the performance of progressive scheduling, since the training in the front stage is totally based on local patches while it has a large learning rate. We report FID comparison 1.66 (stochastic) v.s. 2.05 (progressive) on CelebA-64×64 and 3.11 (stochastic) v.s. 3.85 (progressive) on FFHQ-64×64. Therefore, unless otherwise specified, we employ stochastic patch size scheduling, as discussed in Section 3.2, for the following experiments.

## 4.2 Experiments on Large-scale Dataset

In this section, we aim to compare our Patch Diffusion Model (PDM) with other state-of-the-art diffusion model baselines in terms of both generation quality and training efficiency. As shown in Table 1, Patch Diffusion generally works well in terms of FID while having significantly reduced training time. Note, the FID results of our model and backbone are based on 50 deterministic reverse steps while the other baselines need a much larger number of reverse steps and are much slower in sampling.

We also combine our method with the Latent Diffusion [45] framework to achieve dramatic reduction in training time cost for high-resolution image synthesis, such as LSUN-Bedroom/Church-256×256 [63] and ImageNet-256×256 [9], and we denote it as Latent Patch Diffusion Model (LPDM). We borrow the pretrained image encoder and decoder from Stable Diffusion [45], encode the original images to a smaller latent space and then apply our patch diffusion training. Note the pretrained auto-encoder is not specifically trained for LSUN/ImageNet, which may limit the performance. We use the same latent diffusion model but without patch diffusion training under our codebase as a baseline, and we denote the implementation as LDM-ADM.

For LSUN-Bedroom/Church datasets, we train unconditional sampling, while for ImageNet dataset, we train conditional sampling. Following the classifier-free diffusion guidance (CFG) [14], during training, we randomly drop 10% of the given class labels, and during sampling, we use strength 1.3 for applying CFG, cfg=1.3.

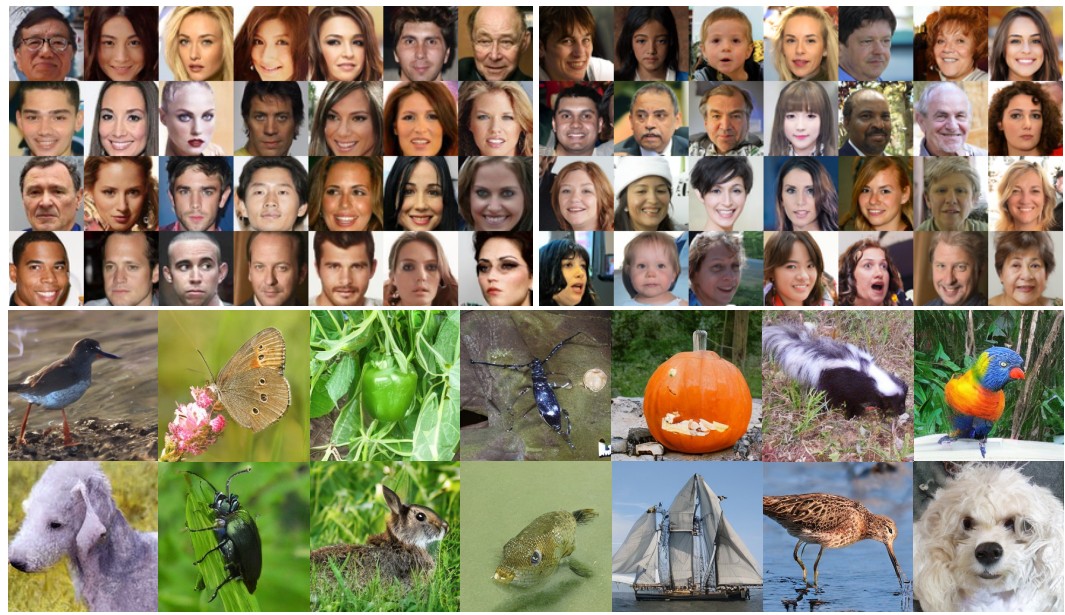

Figure 4: Randomly generated images from Patch Diffusion (EDM-DDPM++ backbone) trained on CelebA-64×64 and FFHQ-64×64, and Latent Patch Diffusion (EDM-ADM backbone) trained on ImageNet-256×256.

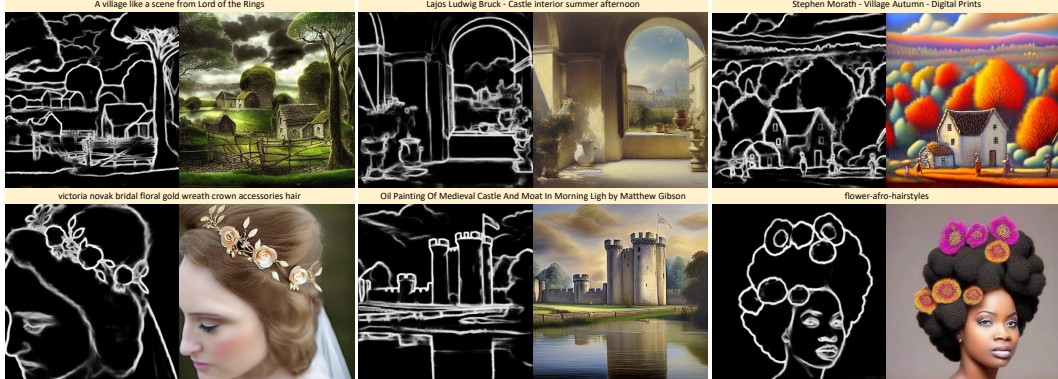

Figure 5: Finetuning Results of ControlNet with Patch Diffusion Training. We finetune a ControlNet on the HED map to image generation task from Stable Diffusion checkpoints with 20k steps.

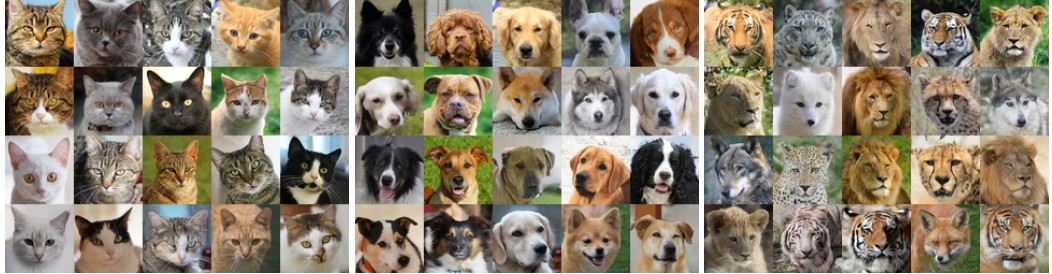

Figure 6: Randomly generated images from Patch Diffusion (EDM-DDPM++ backbone) trained on small datasets, AFHQv2-Cat-64×64, AFHQv2-Dog-64×64, and AFHQv2-Wild-64×64.

We present the FID and training cost in Table 2 and Table 3 for LSUN and ImageNet datasets, repsectively. Patch Diffusion notably surpasses both the baseline and previous state-of-the-art Unet-based diffusion models in both generation quality and training efficiency. We show a number of uncurated image samples generated by Patch Diffusion in Figure 4 and Appendix B. Qualitatively, the images generated by Patch Diffusion exhibit photo-realism and are rich in variety.

Table 1: We report the FID and T-Time (training time in hours) on both CelebA-64×64 and FFHQ-64×64 dataset. For our model and our backbone model EDM [22], we train them on 16xV100 GPUs with batch size 512.

| Algorithm | NFE↓ | CelebA-64×64 | FFHQ-64×64 | Train-Time |
|---|---|---|---|---|
| DDIM [51] | 1000 | 3.51 | - | ∼ 48h |
| DDPM [15] | 1000 | 3.26 | - | ∼ 48h |
| NCSN++ [54] | 1000 | 3.25 | - | ∼ 48h |
| PNDM [33] | 1000 | 2.71 | - | ∼ 48h |
| DDPM++ [23] | 1000 | 1.90 | - | ∼ 48h |
| Soft Diffusion [8] | 300 | 1.85 | - | ∼ 48h |
| EDM-DDPM++ [22] | 50 | **1.66** | **2.60** | ∼ 48h |
| PDM-DDPM++ (ours) | 50 | 1.77 | 3.11 | **∼ 24h** |

Table 2: We report the FID and Train-Time (training time in hours) on LSUN-Bedroom-256×256 and LSUN-Church-256×256 datasets.

| Algorithm | NFE↓ | LSUN-Bedroom | LSUN-Church | Train-Time |
|---|---|---|---|---|
| DDIM [51] | 50 | 10.58 | 6.62 | ∼ 14 days |
| DDPM [15] | 1000 | 7.89 | 4.90 | ∼ 14 days |
| LDM-ADM | 50 | 4.32 | 4.66 | ∼ 8 Days |
| LPDM-ADM (ours) | 50 | **2.75** | **2.66** | **∼ 4 days** |

Table 3: We report the FID, sFID, IS, Precision and Recall on ImageNet-256×256 datasets. We also report the GFLOPs as a measure of compute cost for each algorithm. Each metric is presented in two columns, one without classifier-free guidance and one with, marked with ✗ and ✓, respectively. All methods use NFE 250 steps for reversing the learned diffusion process.

| Evaluation Metric | FID↓ | | sFID↓ | | IS↑ | | Prec.↑ | | Rec.↑ | | Cost |
|---|---|---|---|---|---|---|---|---|---|---|---|
| Classifier-free guidance | ✗ | ✓ | ✗ | ✓ | ✗ | ✓ | ✗ | ✓ | ✗ | ✓ | GFLOPs |
| ADM [10] | 10.94 | 4.59 | 6.02 | 5.25 | 100.98 | 186.70 | 0.69 | 0.82 | 0.63 | 0.52 | 1120 |
| ADM-Upsampled[10] | **7.49** | 3.94 | **5.13** | 6.14 | 127.49 | 215.84 | 0.72 | 0.83 | 0.63 | 0.53 | 742 |
| LDM-8 [45] | 15.51 | 7.76 | - | - | 79.03 | 209.52 | 0.65 | 0.71 | 0.63 | **0.62** | 53 |
| LDM-4 [45] | 10.56 | 3.60 | - | - | 103.49 | **247.67** | **0.84** | **0.87** | 0.35 | 0.48 | 103 |
| LPDM-ADM (ours, cfg=1.3) | 7.64 | **2.72** | 5.36 | **4.86** | **130.23** | 243.25 | 0.73 | 0.84 | **0.63** | 0.57 | **78** |

## 4.3 Experiments on Finetuning

We evaluate Patch Diffusion in finetuning large-scale pretrained diffusion models. We plug it into ControlNet [64] as one example. We use the data proposed by Brooks et al. [3] as our base datasets and extract the HED maps by the HED boundary detector [62] as the input controls. We then finetune the ControlNet on the HED map to image generation task from the Stable Diffusion 'v1-5' checkpoint with 20k steps. We show the qualitative generation from HED maps to images in Figure 5. We observe that patch diffusion can be effectively applied to fine-tuning without compromising performance, while also enhancing training efficiency by approximately two times.

## 4.4 Experiments on Limited-size Dataset

We further investigate whether coordinate-guided score matching could improve the data efficiency of diffusion models. Note by doing random cropping with different patch sizes, even one single image could be extended to thousands of patch samples, which helps overcome the overfitting issue.

Specifically, we conduct experiments on three popular small datasets, AFHQv2-Cat, -Dog, and -Wild, each with as few as around 5k images [7]. We train our Patch Diffusion and the baseline approach EDM-DDPM++ from scratch for a duration of 75 million images. We compare the training cost and generation quality between different methods. As shown in Table 4, we observe that by using patch score matching, our model consistently outperforms the baseline model across all three datasets in terms of FID, while achieving ≥ 2× faster training at the same time. We show a number of uncurated

Table 4: **Results reported on the limited-size dataset (AFHQ-v2).** For all models, we train them on 16xV100 GPUs with batch size 512 for a duration of 75 million images.

| Data($\sim 5k$ images) | Algorithm | FID $\downarrow$ | NFE | Train-Time |
|---|---|---|---|---|
| AFHQv2-Cat | EDM-DDPM++ [22] | 4.60 | 50 | $\sim$ 18h |
| | Patch Diffusion (ours) | **3.11** | 50 | **$\sim$ 9h** |
| AFHQv2-Dog | EDM-DDPM++ [22] | 4.94 | 50 | $\sim$ 18h |
| | Patch Diffusion (ours) | **4.80** | 50 | **$\sim$ 9h** |
| AFHQv2-Wild | EDM-DDPM++ [22] | 2.59 | 50 | $\sim$ 18h |
| | Patch Diffusion (ours) | **1.93** | 50 | **$\sim$ 9h** |

image samples generated by Patch Diffusion in Figure 6. This experiment demonstrates that patch diffusion training could help improve the data efficiency of diffusion models.

## 4.5 Experiments on Image Extrapolation

In this section, we evaluate the out-painting performance of Patch Diffusion on the LSUN-Bedroom dataset. We initially enlarge the coordinate system to a higher resolution, maintaining the range [-1, 1]. The reference image is positioned at the center of the expanded coordinate system and remains static throughout the Patch Diffusion reversal process. In Figure 7, we extend the pixel manifold to dimensions of $384 \times 384$, even though our Patch Diffusion model is trained on $256 \times 256$ image samples. The model effectively generates new content beyond the original boundary. Additionally, we present the extrapolation to a $512 \times 512$ resolution in Appendix D.

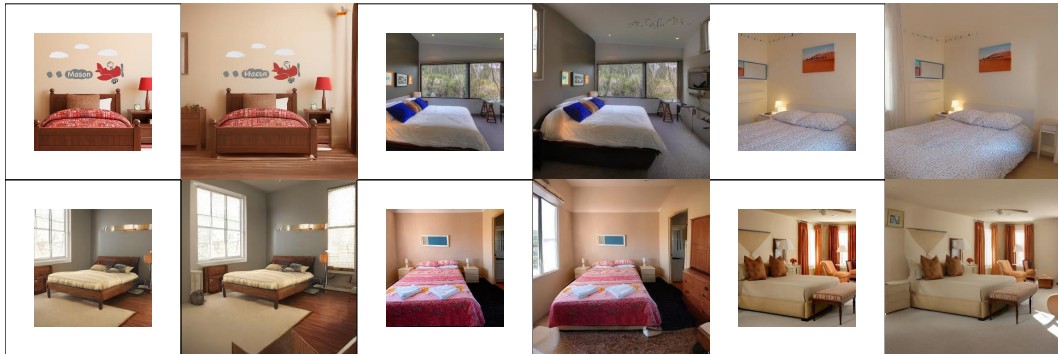

Figure 7: **Extrapolation Results.** Patch Diffusion could generate beyond the boundary by extrapolating the learned coordinate manifold. For each pair of images, the left panel is the reference image in resolution $256 \times 256$ and it is fixed in the center during the reverse process of Patch Diffusion, while the right panel shows the generated sample in resolution $384 \times 384$, where the out-of-boundary region is regenerated. Note our model is trained only on $256 \times 256$ images.

## 5 Conclusion and Future Work

We present Patch Diffusion, a novel patch-level training framework that trains diffusion models via coordinate conditioned score matching. We also propose to diversify the patch sizes for score matching in a progressive or stochastic schedule during training, to capture the cross-region dependency at multiple scales. Sampling with our method is as easy as in the original diffusion model. Patch Diffusion could significantly reduce the training time costs, $e.g.$, $2\times$ faster training, while improving the data efficiency of diffusion models, $e.g.$, improving the performance of diffusion models trained on relatively small datasets. Going forward, the current coordinate system could be further improved by advanced positional embeddings, such as periodic one [38], to better incorporate the position information. We also leave the theoretical proof of the convergence of patch-wise score matching in general cases as our future work.

## Acknowledgments

Z. Wang, H. Zheng, and M. Zhou acknowledge the support of NSF-IIS 2212418, NIH-R37 CA271186, and the NSF AI Institute for Foundations of Machine Learning (IFML).

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

# Appendix

## A  Theoretical Interpretations

In this section, we provide mathematical intuitions for our patch diffusion from two perspectives.

**Markov Random Field.**  Markov Random Field (MRF) has been widely used to represent image distributions [28, 29] due to its compactness and expressiveness in modeling dependence. Usually, images are modeled as an undirected regular graph (pixel grids) $\mathcal{G} = (\mathcal{V}, \mathcal{E})$ in MRF, where each pixel is considered as a graph vertex, and each vertex will connect to its neighboring pixels on the image. Therefore, we can adopt the clique factorization form to represent the PDF defined over images:

$$p(\boldsymbol{x}) = \frac{1}{Z} \prod_{v \in \mathcal{V}} \phi_v(\boldsymbol{x}_v) \prod_{e \in \mathcal{E}} \phi_e(\boldsymbol{x}_e), \tag{6}$$

where $\phi_v$ and $\phi_e$ are node and edge potential functions, $Z$ is a normalization term, $\boldsymbol{x}_v$ and $\boldsymbol{x}_e$ index the random variables via the subscripts $v$ or $e$. One can see $\phi_v$ is a univariate function and $\phi_e$ is a pairwise function.

Next, we point out that the score function of the MRF parameterization can be even neat:

$$\nabla \log p(\boldsymbol{x}) = \sum_{v \in \mathcal{V}} \nabla \log \phi_v(\boldsymbol{x}_v) + \sum_{e \in \mathcal{E}} \nabla \log \phi_e(\boldsymbol{x}_e), \tag{7}$$

where $Z$ is eliminated as it is irrelevant to $\boldsymbol{x}$. As suggested by Equation 7, the score function can be eventually decomposed into independent pieces. That being said, we can separately learn each score function $\nabla \log \phi_v$ and $\nabla \log \phi_e$ first, and then average them up to approximate the entire score function during the inference. Our training procedure can be viewed as: each time we sample patches which contains subsets of $\mathcal{V}$ and $\mathcal{E}$, and we conduct score matching on the corresponding $\nabla \log \phi_v$ and $\nabla \log \phi_e$. Note that $\nabla \log \phi_v$ and $\nabla \log \phi_e$ is not necessarily spatial-invariant. We condition the network on the coordinates to model the location dependence for $\nabla \log \phi_v$ and $\nabla \log \phi_e$.

**Linear Regression.**  We also provide an alternative interpretation for patch-wise score matching through the lens of least square. We consider a multivariate Gaussian distribution as the demonstration example. Suppose we parameterize our target distribution as $p_{\boldsymbol{\mu}}(\boldsymbol{x}) = \mathcal{N}(\boldsymbol{x}|\boldsymbol{\mu}, \boldsymbol{\Sigma})$ where $\boldsymbol{\mu}$ is the optimizee. Then its score function is written as: $\nabla \log p_{\boldsymbol{\mu},\boldsymbol{\Sigma}}(\boldsymbol{x}) = -\boldsymbol{\Sigma}(\boldsymbol{x} - \boldsymbol{\mu})$ [18]. The original score matching with full-size image is equivalent to the following least square problem:

$$\mathbb{E}_{\boldsymbol{x} \sim p(\boldsymbol{x})} \mathbb{E}_{\boldsymbol{\epsilon} \sim \mathcal{N}(\boldsymbol{0}, \sigma_t^2 \boldsymbol{I})} ||\boldsymbol{\Sigma}\boldsymbol{\mu} - (\boldsymbol{\Sigma}\boldsymbol{x} - \boldsymbol{\epsilon}/\sigma_t^2)||_2^2 \tag{8}$$

On the other hand, the patch distribution can be considered as the marginal distribution on a subset of random variables. The transformation between patch distribution and full-size image distribution is a marginalization integral, which is as simple as a linear operator (more precisely an orthogonal projection). In our Gaussian example, the patch distribution has a closed form: $p_{\boldsymbol{\mu}}(\boldsymbol{x}_S) = \mathcal{N}(\boldsymbol{x}|\boldsymbol{P}_S\boldsymbol{\mu}, \boldsymbol{P}_S\boldsymbol{\Sigma}\boldsymbol{P}_S^\top)$, where $S$ indices a set of pixels within an image patch identified by the location and size, and $\boldsymbol{P}_S$ is a selection matrix associated with $S$. Then patch-wise score matching can be written as:

$$\mathbb{E}_S \mathbb{E}_{\boldsymbol{x} \sim p(\boldsymbol{x})} \mathbb{E}_{\boldsymbol{\epsilon} \sim \mathcal{N}(\boldsymbol{0}, \sigma_t^2 \boldsymbol{I})} ||\boldsymbol{P}_S\boldsymbol{\Sigma}\boldsymbol{\mu} - (\boldsymbol{P}_S\boldsymbol{\Sigma}\boldsymbol{x} - \boldsymbol{\epsilon}/\sigma_t^2)||_2^2. \tag{9}$$

From the linear regression perspective, the only difference between patch-based and full-size score matching is the measurement matrix. At first glance, patch diffusion trades the computation cost off the well-posedness. However, we argue that due to well-known redundancy and symmetry in image distributions, recovering the whole image distribution under limited observation can be factually feasible. This guarantees our patch diffusion can also converge to the true distribution. We consider a general argument beyond Gaussian examples as a promising future exploration.

## B  More Generated Images.

More generated images from Patch Diffusion are listed below.

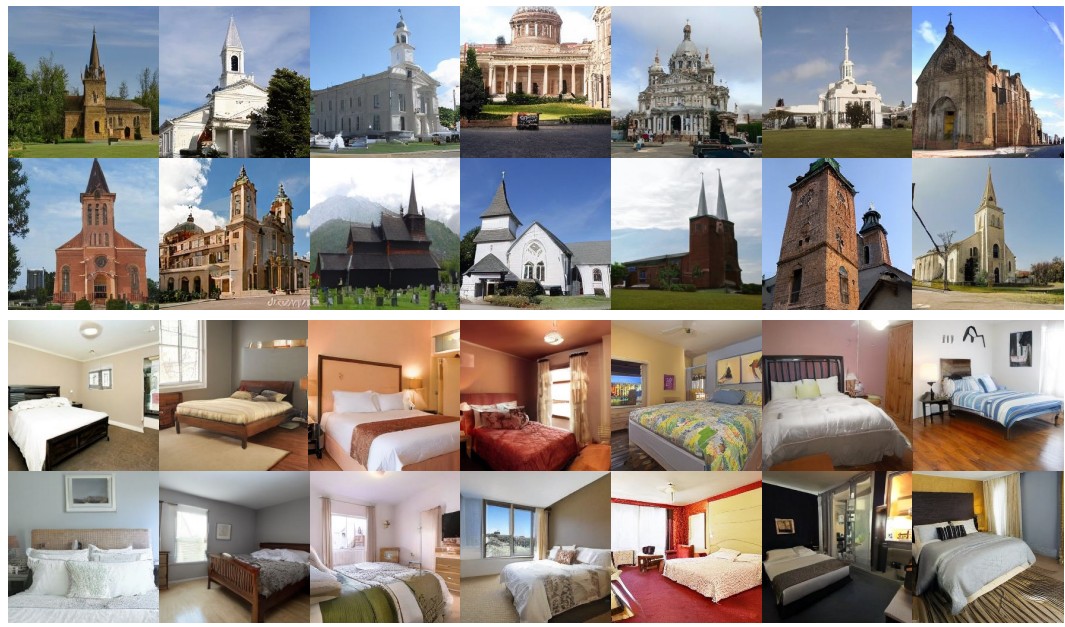

Figure 8: Randomly generated images from Patch Diffusion (EDM-DDPM++ backbone) trained on LSUN-Bedroom/Church-256×256.

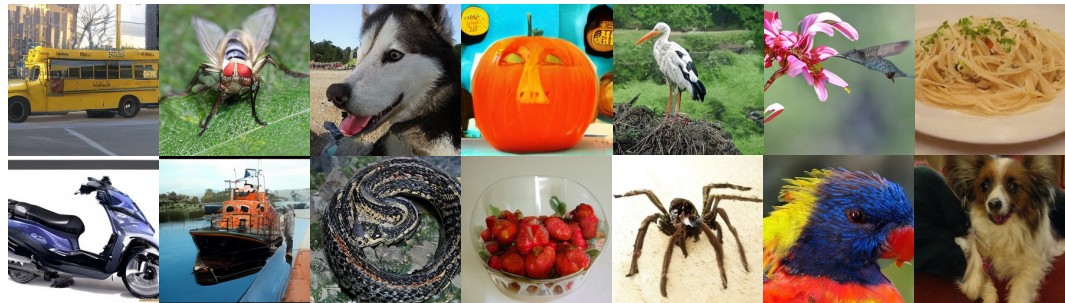

Figure 9: More generated images from Latent Patch Diffusion (EDM-ADM backbone) trained on ImageNet-256×256..

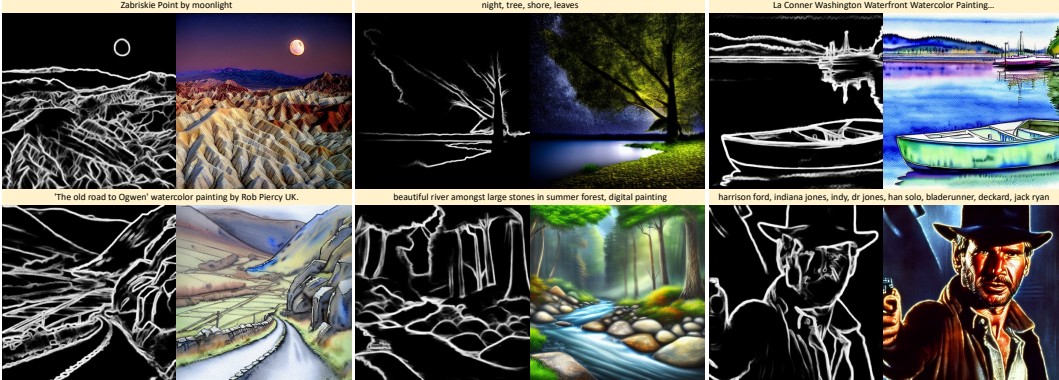

Figure 10: More Finetuning Results of ControlNet with patch diffusion training.

## C   Potential Social Implications

Our work might lead to common negative social impacts of generation models for computer vision. One concern is the proliferation of fake or manipulated images, leading to a crisis of trust and

credibility. As these models become more sophisticated, it turns increasingly difficult to discern between real and generated images, undermining the integrity of visual evidence. This can have significant implications in journalism, forensics, and other fields that heavily rely on accurate visual representation. Image generation models can also be misused for harmful purposes such as creating realistic but false identities, deepfake pornography, or even propaganda and disinformation campaigns. These applications can lead to privacy violations, cyberbullying, defamation, and manipulation of public opinion. Therefore, it is crucial to address the potential negative social impacts of image generation models and implement ethical guidelines and safeguards to mitigate harmful effects.

# D    More Extrapolation Results

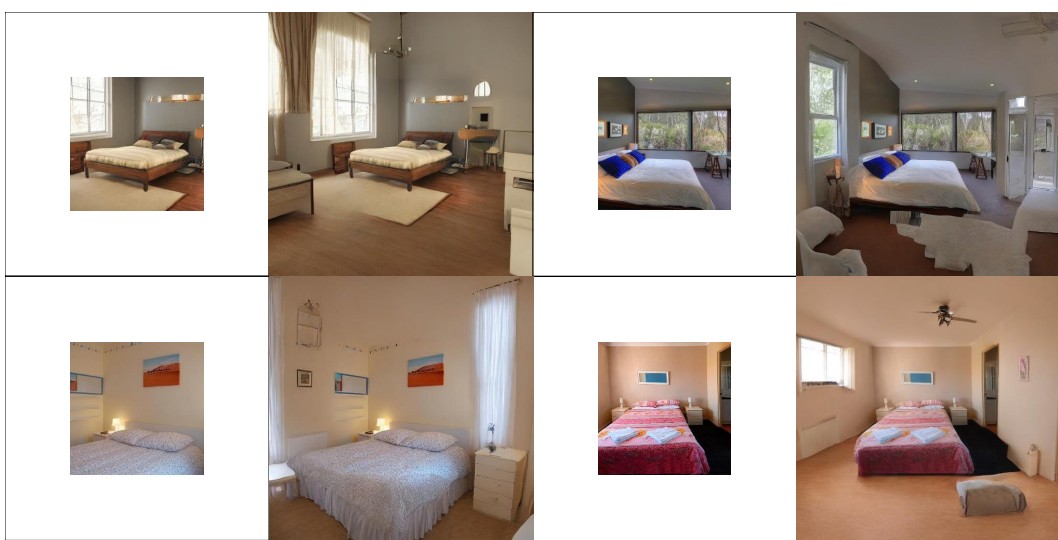

Figure 11: **Extrapolation Results.** Patch Diffusion could generate beyond the boundary by extrapolating the learned coordinate manifold. For each pair of images, the left panel is the reference image in resolution $256 \times 256$ and it is fixed in the center during the reverse process of Patch Diffusion, while the right panel shows the generated sample in resolution $512 \times 512$, where the out-of-boundary region is regenerated. Note our model is trained only on $256 \times 256$ images.

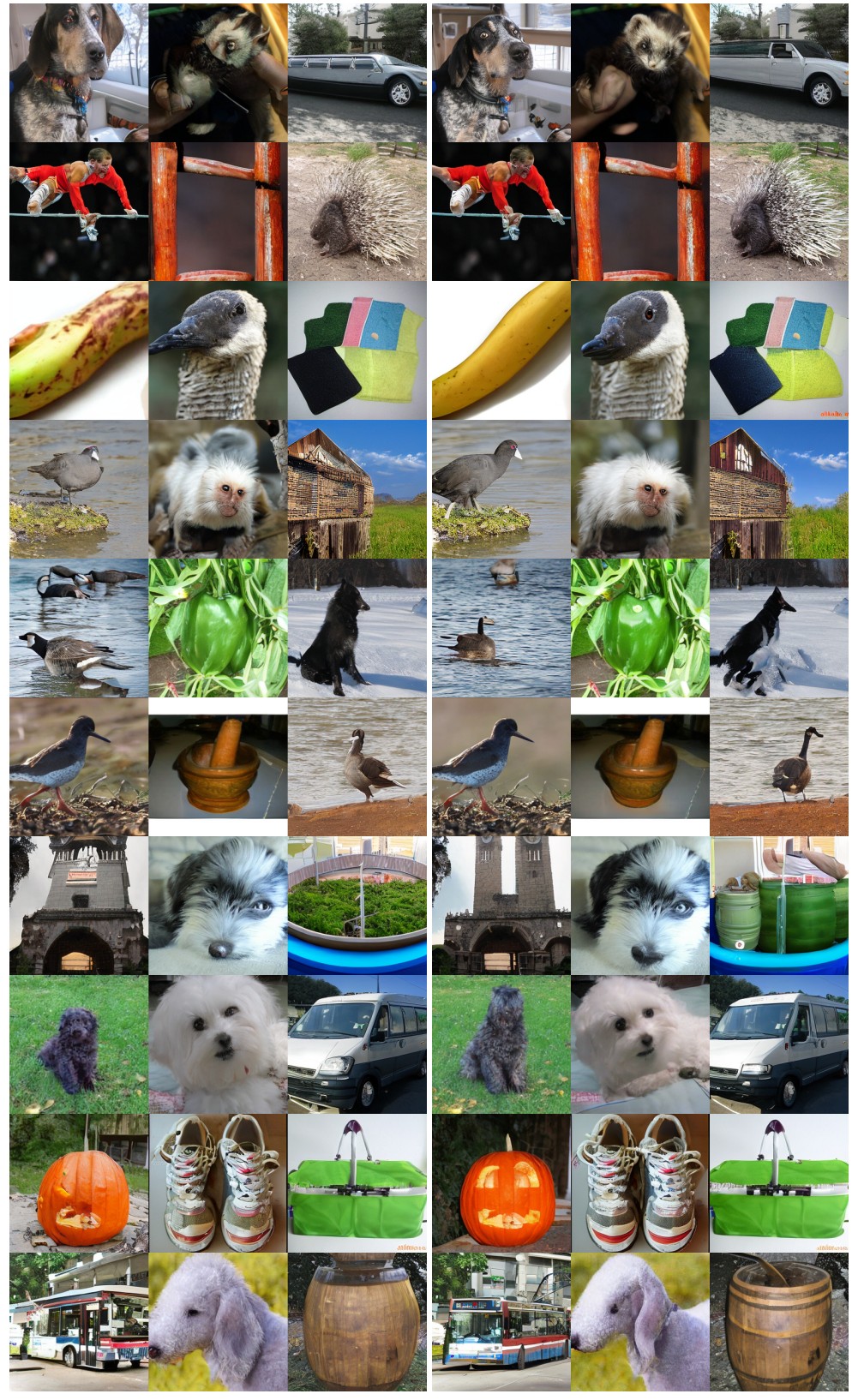

Figure 12: Randomly generated images from Latent Patch Diffusion (EDM-ADM backbone) trained on ImageNet-256×256. Left images are generated without CFG, while the right images are generated with CFG.

