# OpenReview forum: "Patch Diffusion: Faster and More Data-Efficient Training of Diffusion Models"
_NeurIPS.cc/2023/Conference — NeurIPS 2023 poster_

### Official Review · Reviewer_NLEo · 2023-07-02

**Soundness:** 2 fair
**Presentation:** 2 fair
**Contribution:** 2 fair
**Rating:** 5
**Confidence:** 5

**Summary:**

This paper presents PatchDiffusion, a novel framework designed to address the scalability challenges faced by most diffusion models in terms of training and sampling. The proposed framework adopts a patch-wise training approach, where a denoising network is trained on image patches rather than the entire high-resolution images. To generate patches at the target resolution, PatchDiffusion leverages progressive or stochastic scheduling techniques that utilize different patch sizes throughout the training process. Experimental results on a small-scale dataset demonstrate that PatchDiffusion not only enhances the quality of generated samples but also reduces the overall training time. Furthermore, from the experiment results for medium-scale datasets, PatchDiffusion could be a resource-efficient solution for diffusion training methods.

**Strengths:**

This paper tackles a significant challenge in training diffusion models, which often involve substantial computational costs. To address this issue, the authors propose PatchDiffusion as an optional solution that can be seamlessly integrated into any diffusion model pipeline, regardless of the chosen backbone, sampler, or other modules within the pipeline.

I think that PatchDiffusion operates as a form of data augmentation, thereby enhancing the generation quality of the model. This approach provides additional benefits beyond reduced computational costs.

**Weaknesses:**

[Limited experiments - important baseline missing and low performance]

The primary motivation behind patch-wise training is to reduce computation costs during training. Given this motivation, it might be worth considering a more efficient backbone instead of U-Net. Recently, successful approaches have replaced U-Net with ViT, as demonstrated in the papers "Scalable Diffusion Models with Transformers" (arXiv'22) and "All are Worth Words: A ViT Backbone for Diffusion Models" (CVPR'23).

In the case of DiT, reducing the latent resolution by patchifying with a 2x2 patch size has shown improvements in training scalability and performance. It is worth exploring whether applying patch-wise training to DiT and U-ViT backbones could yield better results. However, the potential gains from such an approach are uncertain.

Additionally, it is important to note that in class-conditional image generation on ImageNet-1K, the FID score of PatchDiffusion appears to be significantly worse than that of DiT and other related works. While the current state-of-the-art methods achieve an FID score of less than 4, this work reports a score of around 7.65.

[Writing needs to be improved. ]

The writing quality of the paper could be improved further, especially in terms of highlighting the comparison between this work and the baselines. It would be beneficial to include a figure illustrating the trade-off between FID (or any other measure of generation quality) and FLOPs. This would help in understanding how PatchDiffusion and other efficient diffusion backbones compare to each other. The authors can refer to similar trade-off figures presented in the DiT and U-ViT papers.

Evaluating the quality of writing can be subjective, and I am open to hearing other reviewers' comments on this matter.

**Questions:**

#1. The primary objective of this work is to minimize the computational cost of training/inference through patch-wise training. However, it could be worthwhile to consider an alternative solution by employing a more efficient backbone inherited from ViT, which incorporates a patch-fying (i.e., tokenizing) module. Including a comparison of this work with DiT or U-ViT in the paper would further highlight the benefits of this approach. Moreover, presenting training compute vs. FID plots for the comparison would greatly assist in determining the most effective approach.

#2. In my understanding, the true advantages of this work are likely to be demonstrated through experiments on fine-tuning. This approach has the potential to reduce the fine-tuning cost for any pre-trained diffusion backbone. In this context, incorporating LoRA-type methods could complement this approach. Including empirical analysis of this nature in the paper would enhance the understanding of the benefits of this work.

#3. The GAN community has explored patch-wise training in various ways. Recently, Any-resolution GAN (ECCV’22) has been introduced as a promising solution for training generative models with variable-size images. Although this paper primarily aims to reduce training costs rather than utilizing multiple size images in the training procedure, it would be interesting to explore whether the proposed framework can be extended to generate variable-size images. Doing so would further highlight the benefits of this framework.

#4. A minor comment regarding the inclusion of "Appendix A." To provide a concise overview of the theoretical interpretation of the patch-wise training scheme, it would be beneficial to incorporate a brief summary of these observations within the main body of the paper.

**Limitations:**

The limitations of this work were not explicitly outlined, and I didn’t observe any discussion regarding potential negative societal impacts. However, there is no clear negative societal impact, since all experiments are conducted in controlled benchmark datasets.

---

> ### Author Rebuttal · Authors · 2023-08-07
>
> We thank Reviewer NLEo for providing constructive suggestions. Below, we address each concern raised in your comment point by point. Please let us know if you have any further questions or whether this adequately addresses all these issues.
>
> > Q1+Weakness 1: it could be worthwhile to consider an alternative solution by employing a more efficient backbone, such as DiT. Also, plotting training compute vs. FID for the comparison would greatly assist in determining the most effective approach. ...
>
> We acknowledge that both U-Net based and transformer-based diffusion models have attracted significant attention and exhibited strong performance. As of now, there is no conclusive evidence favoring one over the other. Notably, U-Net based diffusion models continue to dominate text-to-image generation, as evidenced by their usage in Stable Diffusion, Imagen, and DALL-E 2.
>
> In reference to your statement, "Given this motivation, it might be worth considering a more efficient backbone inherited from ViT," we respectfully present a differing perspective. Quoting the DiT paper, "Figure 2 (right) demonstrates the computational efficiency of DiT-XL/2 (**118.6 Gflops**) relative to latent space U-Net models like LDM-4 (**103.6 Gflops**) and notably more efficient than pixel space U-Net models like ADM (1120 Gflops) or ADM-U (742 Gflops)." In the context of being applied to latent space, U-Net proves to be more efficient than Transformer architecture. In our ImageNet-1K experiment, Patch Diffusion employs ADM-U on latent space (1x4x32x32) with a considerably lower computational cost of around **27 Gflops**, demonstrating its efficiency. It is pertinent to note that our model possesses approximately 290M parameters, whereas DiT-XL/2 boasts 675M parameters. As elucidated in Table 4 of the DiT paper, DiT model exhibits notably inferior performance on ImageNet-1K if it takes less than 300M parameters.
>
> Moreover, transformer-based diffusion models typically necessitate a substantially higher number of training steps to converge. Table 4 of the DiT paper exemplifies this, as DiT-XL/2 requires 7M steps to achieve convergence, while the U-Net model utilized in our work reaches convergence within 0.61M steps.
>
> > Weakness 2: the current state-of-the-art methods (DiT) achieve an FID score of less than 4, this work reports a score of around 7.65.
>
> The performance reported in Table 2 of our manuscript is under the setting of **NOT** adopting classifier-free guidance. Under the same setting, DiT only achieves 9.62 FID score while Patch Diffusion achieves 7.65 FID score, demonstrating the effectiveness of our proposed method. If applying classifier-free guidance to Patch Diffusion, our model could reach 2.74 FID score on ImageNet-1K, which matches the state-of-the-art performance but costs a significantly lower computational cost (27 Gflops v.s. 118.6 Gflops).
>
> > Weakness 3: It is worth exploring whether applying patch-wise training to DiT and U-ViT backbones could yield better results.
>
> Our current Patch Diffusion is built and designed for U-Net based diffusion models. We agree with Reviewer NLEo that applying patch diffusion training onto transformer based diffusion models is also worth further investigation, but that currently is out of the scope of this paper, and we left that for future study.
>
> > Q2: incorporating LoRA for fine-tuning
>
> We appreciate Reviewer NLEo for recognizing the potential applicability of our method in finetuning scenarios. We do agree that our patch diffusion training approach harmonizes effectively with LoRA-type methods, thereby potentially enhancing finetuning efficiency. While LoRA is not the primary focus of this paper, we are open to the prospect of integrating LoRA in future investigations.
>
> > Q3: variable-size images?
>
> Any-resolution GAN [5] is primarily tailored for mixed-resolution datasets from images in the wild, whereas our work focuses on improving the training efficiency of diffusion models. In terms of generating variable-size images, we provided the extrapolation results of Patch Diffusion in the pdf of Response to All.
>
> > Q4: Moving Appendix to the main manuscript.
>
> We would love to include theoretical interpretation in our main paper. We will consider adding a summary of the theoretical interpretation based on the page limit.
>
> > Limitations discussion is missing.
>
> We discussed limitations and potential future work in Section 5, and potential negative social impacts in Appendix B.1.

---

> > ### Author Response · Authors · 2023-08-18
> >
> > We would greatly appreciate it if you could review our response by August 21st. After that date, it might be challenging for us to engage in further discussions. If you have any follow-up questions, please don't hesitate to reach out. We deeply value your expertise and time.

---

> > > ### Comment · Reviewer_NLEo · 2023-08-21
> > > **Increasing my score from BR to BA.**
> > >
> > > Thank you for your detailed response. It has clarified most of the concerns I initially had about the paper. I had misunderstood certain aspects, particularly regarding the FiD score of DiT. Beyond the clarifications, I find the extrapolation (out-painting) results particularly interesting, since it's very practical that this diffusion model, trained using patch-wise methods, inherently offers the out-painting capability. As a result, I have revised my evaluation to BA (borderline accept).
> > >
> > > Regarding the manuscript, I agree with Reviewer Vqdw's observations about the subpar quality of the main figure. I recommend updating both the main figure and the manuscript itself.

---

> > > > ### Author Response · Authors · 2023-08-21
> > > >
> > > > We are elated to have addressed your concerns regarding our paper. Your insights, particularly about the extrapolation results, are highly valuable to us. Rest assured, we are committed to refining the clarity and quality of the main figure and to expanding our discussion on transformer-based diffusion models, such as DiT.

---

### Official Review · Reviewer_Vqdw · 2023-07-05

**Soundness:** 3 good
**Presentation:** 3 good
**Contribution:** 2 fair
**Rating:** 6
**Confidence:** 5

**Summary:**

This paper presents a new training technique that improves the training speed of diffusion models. Instead of training the diffusion model on the entire image, the authors propose training on sampled patches of the image. This approach maintains the theoretical foundation of the diffusion model by keeping the training objective function mostly unchanged. By combining this approach with a fully-convolutional U-Net architecture, the computational complexity is reduced, resulting in faster training. To minimize the quality difference between the partial image approach and the conventional whole image approach, a stochastic/progressive patch size scheduling was proposed, and the corresponding ablation study was conducted. This study investigates the optimal probability of using the whole image, considering the trade-off between training time and generation quality. Summarizing, the proposed method enables faster learning while preserving the theoretical foundation and generation quality of traditional diffusion models.






**Strengths:**

This paper presents two significant advantages of training diffusion models using partial images instead of the entire images. Firstly, it reduces model complexity, leading to faster training, which is especially beneficial for state-of-the-art baseline diffusion models that require extensive GPU hours for training. This approach offers the potential for energy-efficient training by effectively reducing the overall training time.

Secondly, training with partial images proves effective in scenarios with limited datasets, outperforming traditional methods. In cases where the dataset is insufficient, diffusion models struggle to accurately predict the true data distribution due to overfitting (limited data is essentially a sparse sampling of true data distribution). By partitioning the image into patch units, the training process simulates a larger dataset, providing the model with more training samples. This enables the diffusion model to estimate a more accurate data distribution, resulting in improved generation quality.

The paper conducted a series of experiments encompassing large-scale datasets, limited-size datasets, and fine-tuning scenarios, to demonstrate the effectiveness of training images in patch units. This approach maintains performance while significantly improving training efficiency.

**Weaknesses:**

The key idea of this paper is to modify the input data format while maintaining the training process of the diffusion model. This approach aligns with similar strategies proposed in previous works like COCO-GAN. Considering the large overlap in ideas, the authors should present various case studies, such as showcasing the application of this technique to diffusion models, in order to compensate for the limited novelty of the paper.

Firstly, there is a lack of analysis concerning spatial conditions. The authors convert the traditional three-channel format (R, G, B) to a five-channel format (R, G, B, i, j) incorporating location information. Meanwhile, other methods such as positional encoding in modern Transformer structures or Fourier feature methods in Alias-Free GAN have been proposed for spatial conditioning. Multiple experiments in various literature have demonstrated that positional encoding is more effective than raw methods like (i, j). Conducting an ablation study on different spatial conditioning methods and providing an analysis of the most suitable conditioning approach in a diffusion setting would strengthen the paper's credibility.

Secondly, there is insufficient analysis regarding the ability of patch diffusion to generate structural diversity. The experimental results in Table 1 and Table 2 show a notable improvement in the quality of patch diffusion for datasets with weak common structures (e.g., Bedroom, Church) compared to those with strong common structures (e.g., FFHQ, CelebA). Including an analysis of the factors contributing to this quality improvement, such as visualizing attention layers for patch images, would help readers' understanding of the method.

**Questions:**

1. Patch-wise generation:
Is it feasible to generate the entire image by combining generated parts instead of generating it all at once?

2. Patch diffusion for image extrapolation:
What would happen if channels for (i, j) were provided with a range of (-1.2, 1.2)? Can patch diffusion extrapolate the image using this approach?

3. Figure quality:
To enhance the readability and visual appeal of Figure 1, I recommend refining its quality. Currently, the overall figure appears hastily created, resembling a PowerPoint slide. I kindly request aligning each object for the camera-ready submission to improve its structure. The objects currently occupy space without significant value. Also, the range of i, j is -1~1, but the crop has a different scale, such as 16x16. Please provide an example of the actual value when a 16x16 crop is performed. Furthermore, the font size in the figure is very small, making it difficult to read. Considering the available white space, increasing the font size throughout would greatly improve legibility.

4. Typo:
Line 86: "e.g.,." should be corrected to "e.g."

**Limitations:**

The authors adequately addressed the limitations and potential negative societal impact.

---

> ### Author Rebuttal · Authors · 2023-08-07
>
> We thank Reviewer Vqdw for providing the positive feedback and constructive suggestions. We address your questions and provide more clarifications below.
>
> > Firstly, there is a lack of analysis concerning spatial conditions.
>
> Thanks for pointing this out. We agree proper positional embedding could potentially improve the current Patch Diffusion Training. We have tried Fourier Positional Encoding (used in Neural Radiance Fields [1]) as an alternative way to replace the raw coordinates inputs, but we did not see obvious improvement in the preliminary experiment. We will try more in the future (as we mentioned in the future work).
>
> [1] Mildenhall, Ben, et al. "Nerf: Representing scenes as neural radiance fields for view synthesis." Communications of the ACM 65.1 (2021): 99-106
>
> > Secondly, there is insufficient analysis regarding the ability of patch diffusion to generate structural diversity.
>
> We agree with the observation that Patch Diffusion training helps more for datasets with weak common structures, such as Bedroom, Church, and ImageNet (with classifier free guidance, now we reach FID **2.74**). We hypothesize that training on local patches could help to capture more local details in the image and then improve the general generation quality.
>
> > Question 1
>
> It is feasible to do generation by parts. The results will depend on the generation scheme. For example, for the CelebA dataset, if you generate the image with four non-overlapping parts (left-top, right-top, left-bottom, right-bottom), then the model will generate a face with each part independently generated. If the entire coordinate manifold is the input, then the entire image is generated and is consistent.
>
> > Question 2
>
> Thanks for pointing this out. We have provided the extrapolation results in the pdf of **Response to All**.
>
> > Question 3
>
> We thank you for pointing out suggestions in refining Figure 1. We will follow your suggestions and modify it in our next revision.

---

> > ### Comment · Reviewer_Vqdw · 2023-08-18
> >
> > Thank you for the authors' response.
> >
> > For the weakness 1, providing sFID may give some answer. Have you tried to measure sFID? Other diffusion model literatures usually report FID, sFID, precision and recall. Could you provide all other metric scores?
> >
> > For the question 1, stitching multiple images into one can cause seam in the images. Do the Patch Diffusion also suffers from seam?

---

> > > ### Author Response · Authors · 2023-08-19
> > >
> > > We thank Reviewer Vqdw for reviewing the response and providing feedback. We answer your two remaining questions below.
> > >
> > > - In our initial experiment, we assessed the FID on CelebA-64x64 both with and without the NeRF positional embedding. The FID values were 1.81 for Patch Diffusion with NeRF and 1.77 for Patch Diffusion using current coordinates. A comprehensive discussion on NeRF and other metrics will be presented in our subsequent revision.
> > >
> > > - As previously noted, the generative outcome is contingent upon the input coordinates.  If four non-overlapping coordinates are given, the model will independently generate four non-overlapping patches accordingly. A naive combination of these patches will result in seam discrepancies. This outcome is anticipated, given that the input comprises only non-overlapping coordinate sets, preventing the model from achieving coherence in unprovided segments. However, when provided with a complete coordinate manifold, the model can render a unified and coherent image. Patch diffusion can potentially be combined with dedicated methods developed for stitching different region generations, such as overlapping certain regions of different generations [1].
> > >
> > > [1] Bar-Tal, Omer, et al. "Multidiffusion: Fusing diffusion paths for controlled image generation." (ICML 2023).

---

> > > > ### Comment · Reviewer_Vqdw · 2023-08-20
> > > >
> > > > Thanks for the answers.
> > > >
> > > > Actually, our questions for sFID is different from the results of FID. Nevertheless, I think the current evaluation shows a clear advantage. I hope that the authors consider adding sFID evaluations in their final version.
> > > >
> > > > We have read reviews from other reviewers and the authors' responses to each reviewer. I think the authors provide reasonable answers to each comment and their answers to our questions also properly address our concerns. So, I keep my original rating, weak accept.
> > > >
> > > > - Regarding ViT-based backbone experiment (by NLEo)
> > > >
> > > > Considering UNet-based architecture is mainstream for diffusion models, 1) the effectiveness of the UNet-based model is meaningful, and evaluations across different model architectures seem out-of-scope for this paper. Besides, 2) the rebuttal period is insufficient for a Vit-based experiment. (in fact, cvpr'23 is more like concurrent work. I personally do not agree the comparison with the concurrent work is the authors' responsibility.)

---

> > > > > ### Author Response · Authors · 2023-08-20
> > > > >
> > > > > Based on your suggestions, we commit to incorporating the sFID evaluations in our next revision. We thank you for taking the time to thoroughly read all the reviews and acknowledge our contributions, particularly in the realm of UNet-based architecture. Your recognition of the meaningfulness of our Patch Diffusion deeply encourages us. Your comprehensive review and constructive feedback greatly enhance our work.

---

### Official Review · Reviewer_ExTh · 2023-07-07

**Soundness:** 4 excellent
**Presentation:** 4 excellent
**Contribution:** 3 good
**Rating:** 6
**Confidence:** 4

**Summary:**

The paper introduces a path-wise diffusion algorithm for faster training. The authors propose patch coordinate conditioned diffusion models and present a patch-size conditioning scheduling technique for efficient training. The method has similar motivation with patch based GAN such as COCOGAN, but it is applied to diffusion model.

**Strengths:**

The method presented in the paper is simple yet effective, successfully reducing the training time by half. The motivation behind the approach aligns with that of COCOGAN, and it can be seen as a reasonable extension for the diffusion model to reduce computational requirements.

**Weaknesses:**

To learn global structure, the algorithm still need a portion of full-resolution diffusion which introduces a bottleneck in terms of time and memory costs. Additionally, the patch-size scheduling approach appears to be manually designed, potentially limiting its adaptability and automation in optimizing the training process. More controlled experiments on various positional encoding or different hyperparameters would provide better intuition about the work.

**Questions:**

1. Is it possible to extrapolate an images as shown in COCOGAN?
2. What is FID scores of baseline model with same train-time without patch training? (e.g. 24h FIDs of EMM-DDPM++ for CelebA)


**Limitations:**

The author provides some limitations on conclusion section.

---

> ### Author Rebuttal · Authors · 2023-08-07
>
> We thank Reviewer ExTh for providing positive feedback. We address your questions and provide more clarifications below.
>
> > Weakness
>
> The patch-size scheduling could be flexibly set as other values. Different scheduling will induce different levels of gain in training efficiency. The setting shown in the paper is not cherry-picked and is just what we used. We agree incorporating more advanced positional embedding is worth further investigation. We have tried Fourier Positional Encoding (used in Neural Radiance Fields [1]) as a simplest way to replace the raw coordinates inputs, but we did not see obvious improvement in the preliminary experiment. We will try more in the future (as we mentioned in the future work).
>
> [1] Mildenhall, Ben, et al. "Nerf: Representing scenes as neural radiance fields for view synthesis." Communications of the ACM 65.1 (2021): 99-106
>
> > Is it possible to extrapolate an images as shown in COCOGAN?
>
> Thanks for pointing this out. We have provided the extrapolation results in the pdf of **Response to All**.
>
> > What is FID scores of baseline model with same train-time without patch training? (e.g. 24h FIDs of EMM-DDPM++ for CelebA)
>
> Our training logs show that when the EDM baseline method is trained with ~24h, it produces 1.85 FID score on CelebA and 3.36 FID score on FFHQ, respectively.

---

> > ### Comment · Reviewer_ExTh · 2023-08-17
> >
> > Thank you for providing the author response. It seems the model can extrapolate well. After reading the response, I keep my original rate.

---

> > > ### Author Response · Authors · 2023-08-20
> > >
> > > We thank you for your consideration and affirmation after reviewing our response; we genuinely appreciate it.

---

### Official Review · Reviewer_qnyP · 2023-07-09

**Soundness:** 3 good
**Presentation:** 3 good
**Contribution:** 3 good
**Rating:** 6
**Confidence:** 4

**Summary:**

The authors propose a new formulation of training diffusion models by sampling different-sized patches from the training data. The models trained with this formulation have comparable FID scores to models trained on full images on many datasets.

**Strengths:**

The authors proposed a way to train faster diffusion models, which can cut down training time in half with almost similar performance.

Well-written and to the point paper.


**Weaknesses:**

Training with patches of image with same guidance might not work when the training data is more heterogeneous like LAION. In datasets where the subject might not be centered in image, or occupy small portion of the image or existence of multiple object in the scene, I wonder if model can still perform well.

**Questions:**

Do you have any results on models trained on LAION or MS COCO?

**Limitations:**

Yes

---

> ### Author Rebuttal · Authors · 2023-08-07
>
> We thank reviewer qnyP for providing positive feedback and valuable suggestions. Currently, the pretraining scope of our experiment is limited to class conditional experiments due to the limit of computational resources. Thanks for pointing out interesting and promising potentials. We will consider training on text-guided image generation datasets such as LAION or MS-COCO in the future.

---

> > ### Comment · Reviewer_qnyP · 2023-08-15
> > **Thank you**
> >
> > Thank you for the response. I stick to my current score.

---

> > > ### Author Response · Authors · 2023-08-17
> > >
> > > We appreciate the time you've taken to review our work and provide feedback. We are committed to continuous improvement and value your insights.

---

### Official Review · Reviewer_nnk4 · 2023-08-01

**Soundness:** 2 fair
**Presentation:** 3 good
**Contribution:** 3 good
**Rating:** 7
**Confidence:** 3

**Summary:**

The paper proposes a novel training framework for diffusion models, that significantly reduces the training time, while improving data efficiency. For the first time, the proposed method suggests patch-wise diffusion training, which can be deployed to any UNet-based diffusion models. Experimental results show that the patch-wise diffusion training can halve the training time while maintaining comparable or better image quality than the baseline models. On small scale datasets, the proposed method outperformed other baselines, validating the data efficiency of path-wise training.

**Strengths:**

- The paper is well organized and easy to follow. The motivation of the paper is very clear, which is to shorten the training time of diffusion models while maintaining image generation quality.

- Proper ablation studies are delivered to fully validate roles of different components of the model, and affect of different parameters.

- 2x speedup of training time is non-negligible, considering long training time of conventional diffusion models.

- The data efficiency followed by patch-wise training framework is considered a significant discovery.

- The proposed method can be applied to any UNet-based diffusion models in a plug and play method.

**Weaknesses:**

- Albeit the empirical evidences, the theoretical proof of convergence of patch-wise score matching is missing.

- Experiments on high resolution image synthesis, beyond 256x256 resolution is missing. According to Table1 and 2, compared to the baseline method, the proposed patch diffusion showed better performance boost on the larger scale datasets (LSUN-Bedroom&Church), than the smaller scale datasets (CelebaA and FFHQ). Thus, it might imply that the patch diffusion can be more beneficial in high resolution image synthesis scenarios. Therefore, validation on high resolution image datasets, beyond 256x256 resolution could be interesting.


**Questions:**

- While training, when entering the denoiser (UNet), has the small patches resized into the original image size? If not, will there be a difference between using resizing and not?


**Limitations:**

Limitations are addressed by the authors.

---

> ### Author Rebuttal · Authors · 2023-08-07
>
> We thank Reviewer nnk4 for your positive and valuable feedback. We appreciate the time and effort you've taken to review our work. We have carefully considered your comments and suggestions, and we would like to respond to each of them.
>
> > Albeit the empirical evidence, the theoretical proof of convergence of patch-wise score matching is missing.
>
> Thanks for pointing this out. We mentioned in Section 5 that providing theoretical proof of convergence for patch-wise score matching is a potential future work. Currently, we provide a theoretical interpretation for Patch Diffusion in Appendix A.
>
> > Experiments on high resolution image synthesis, beyond 256x256 resolution is missing.
>
> Thanks for the suggestion! Training on high-resolution images is still computationally expensive for us due to our limited computational resources. We will consider training on 512x512 images in the future. We also provide image extrapolation results in the reponse to all, which shows that our model could generate larger size images, such as 384x384 and 512x512 images, while only trained on 256x256 images.
>
> > While training, when entering the denoiser (UNet), has the small patches resized into the original image size? If not, will there be a difference between using resizing and not?
>
> No, the small patches are not resized into the original image sizes when entering the UNet. Using small patches for training could significantly improve the training efficiency.

---

### Author Rebuttal · Authors · 2023-08-07

# Response to All

We'd like to thank all five reviewers for their insightful comments and suggestions. We hereby provide the image extrapolation results that Reviewer ExTh, Vqdw and NELo are interested in, and the state-of-the-art ImageNet-1K(256x256) FID **2.74** for Patch Latent Diffusion with the use of classifier-free guidance. We will incorporate these new results and improvements into the camera ready version of the paper.

---

### Decision · Program_Chairs · 2023-09-21

**Decision:**

Accept (poster)

**Comment:**

This paper proposes a new method for training diffusion models on patches, thereby speeding up the training process as compared to when training on only full images. Experiments show that the proposed method is promising. The reviewers assess this work as an substantial contribution to an important problem. There was substantial discussion between the authors and reviewers, resulting in a unanimous vote to accept from the reviewers. As such I'm happy to recommend accepting this work.